# Avian influenza A virus susceptibility, infection, transmission, and antibody kinetics in European starlings

**Jeremy W. Ellis, J. Jeffrey Root, Loredana M. McCurdy, Kevin T. Bentler, Nicole L. Barrett, Kaci K. VanDalen¤a, Katherine L. Dirsmith¤b, Susan A. Shriner***

National Wildlife Research Center—Wildlife Services, Animal Plant Health Inspection Service, United States Department of Agriculture, Fort Collins, Colorado, United States of America

¤a Current address: NIH/NIAID Office of Biodefense Research and Surety, Rockville, Maryland, United States of America
¤b Current address: Field Operations District 1—Veterinary Services, Animal Plant Health Inspection Service, United States Department of Agriculture, San Juan, Puerto Rico, United States of America
* Susan.A.Shriner@usda.gov

**Data Availability Statement:** All relevant data are within the manuscript and its Supporting Information files.

## Abstract

Avian influenza A viruses (IAVs) pose risks to public, agricultural, and wildlife health. Bridge hosts are spillover hosts that share habitat with both maintenance hosts (e.g., mallards) and target hosts (e.g., poultry). We conducted a comprehensive assessment of European starlings (*Sturnus vulgaris*), a common visitor to both urban and agricultural environments, to assess whether this species might act as a potential maintenance or bridge host for IAVs. First, we experimentally inoculated starlings with a wild bird IAV to investigate susceptibility and replication kinetics. Next, we evaluated whether IAV might spill over to starlings from sharing resources with a widespread IAV reservoir host. We accomplished this using a specially designed transmission cage to simulate natural environmental transmission by exposing starlings to water shared with IAV-infected mallards (*Anas platyrhynchos*). We then conducted a contact study to assess intraspecies transmission between starlings. In the initial experimental infection study, all inoculated starlings shed viral RNA and seroconverted. All starlings in the transmission study became infected and shed RNA at similar levels. All but one of these birds seroconverted, but detectable antibodies were relatively transient, falling to negative levels in a majority of birds by 59 days post contact. None of the contact starlings in the intraspecies transmission experiment became infected. In summary, we demonstrated that starlings may have the potential to act as IAV bridge hosts if they share water with IAV-infected waterfowl. However, starlings are unlikely to act as maintenance hosts due to limited, if any, intraspecies transmission. In addition, starlings have a relatively brief antibody response which should be considered when interpreting serology from field samples. Further study is needed to evaluate the potential for transmission from starlings to poultry, a possibility enhanced by starling's behavioral trait of forming very large flocks which can descend on poultry facilities when natural resources are scarce.

**Funding:** This work was supported by the U.S. Department of Agriculture, Animal and Plant Health Inspection Service. The funders had no role in study design, data collection and analysis, decision to publish, or preparation of the manuscript.

**Competing interests:** The authors have declared that no competing interests exist.

## Author summary

Besides causing seasonal influenza, influenza A viruses (IAVs) are important because they can become pathogenic and threaten human, livestock, or wildlife health. Wild birds are the primary reservoir of IAVs which are generally low pathogenic, but when wild bird viruses spill over into poultry, they can evolve to be highly pathogenic to poultry and sometimes to wild birds or humans. Thus, understanding how viruses move from wild birds into poultry is important. Aquatic birds such as ducks and geese are commonly infected with IAVs, but in many regions, these birds are uncommon on farms. Therefore, species that use both aquatic and agricultural areas may pose a risk by moving IAVs from aquatic birds to poultry. In this paper we evaluated whether European starlings, a species commonly found in both aquatic and agricultural habitats, can be infected by sharing water with IAV-infected ducks. We found that starlings can become infected when exposed to contaminated water, but IAV does not readily transmit between starlings. Consequently, starlings may pose a risk for spillover of IAVs to farms but are unlikely to maintain infections without exposure to other species.

## Introduction

Influenza A viruses (IAVs) pose a threat to both public and agricultural health when high consequence strains spread in human and livestock populations. For example, in 2015 the United States (US) experienced multiple large scale poultry outbreaks after a highly pathogenic H5 Eurasian strain (clade 2.3.4.4) IAV was introduced to North America [1] and spread widely across the nation in wild, captive, and domestic birds [2,3]. Nearly 50 million poultry died or were euthanized as a result, with an estimated economic loss of $1.6 billion (US) to the US poultry industry [4,5]. Moreover, in China multiple avian IAVs (e.g., H5N1, H7N9) associated with high human case fatality rates have emerged and spilled over into humans [6–8]. Reducing the risk to public and agricultural health posed by emerging and re-emerging IAVs requires rigorous assessments of potential transmission pathways between wild bird reservoir species and spillover hosts.

Recent IAV outbreaks in poultry have prompted multiple epidemiologic investigations designed to identify potential transmission pathways and associated risk factors [9–14]. A case control study of the 2015 outbreaks in egg layer farms in the US found that outbreak farms were more likely to report the presence of wild waterfowl and shorebirds in nearby fields compared to uninfected farms [13]. Similarly, a study of US turkey farms during the same epizootic found that wild birds were observed in turkey barns on a third of affected farms [14]. Farm managers reported observing starlings and sparrows in poultry barns, prompting the authors to suggest that small perching birds could be important in the initial introduction of IAVs into commercial poultry.

Molecular epidemiology studies have also investigated wild bird involvement in poultry outbreaks. In a US study, researchers suggested that while IAV spread was likely human-mediated, wild birds may have been responsible for initial introductions [12]. Similarly, several molecular epidemiology studies of European outbreaks have identified wild bird presence to be a risk factor for introduction of IAVs to poultry. In one study, researchers found that wild birds were likely responsible for the introduction of outbreak viruses [10]. A second study found that indirect introduction of IAVs from material contaminated by wild birds was the most likely transmission pathway for some farms and direct contact with wild birds was the likely pathway on other farms, especially for those with outdoor holdings [11]. A German

study found that poultry density was a risk factor for farm spread, but also found that early in the epidemic direct and indirect contact with infected wild birds was a primary risk factor for farms with outdoor birds [15].

Across these studies, wild birds were regularly identified as the initial source of outbreak IAVs, with subsequent farm to farm transmission associated with known biosecurity risks (e.g., equipment sharing, service visits) or farm characteristics (e.g., high density poultry areas, proximity to other cases, or proximity to wild bird usage areas). IAV maintenance hosts such as wild waterfowl and shorebirds clearly pose a spillover threat to poultry farms, especially for operations with outdoor holdings. However, in most areas waterfowl and shorebirds are absent or infrequently observed on large-scale commercial operations [16,17]. Thus, synanthropic species that are commonly observed sharing both aquatic and farm habitats may act as bridge hosts that facilitate spillover from aquatic bird maintenance hosts to poultry [18]. While biosecurity guidelines are generally available for reducing human-mediated risks on farms, less attention has focused on assessments of wild bird incursion risks [19], especially for common passerines. In a previous field investigation of wildlife at an IAV outbreak site in the US, we found evidence of a possible H5 IAV infection in a European starling (*Sturnus vulgaris*) [19]. That finding motivated the evaluation of starlings as IAV maintenance or bridge hosts described herein. Starlings are a common passerine at the wildlife-agricultural interface, but only a handful of studies have examined the role these birds might play in IAV poultry outbreaks [20].

The objective of this study was to conduct a comprehensive evaluation of IAV in European starlings by assessing susceptibility, infection dynamics, environmental transmission, intraspecific transmission, and long-term antibody kinetics. Rigorous evaluations of within host dynamics and realistic transmission scenarios are critical for characterizing pathogen host range, identifying transmission pathways, and providing quantitative data to support risk assessments of viral emergence [21,22]. In this study we used a wild bird IAV subtype (H4N6) that requires a lower level of biocontainment as a surrogate for low pathogenic subtypes (H5s/H7s) that are associated with emergence of highly pathogenic strains of IAV in poultry. While there is no evidence that H4 IAVs cause a significant threat to wildlife, poultry, or human health, the infection kinetics of these viruses are generally similar to low pathogenic H5/H7 strains (e.g., compare infection dynamics in [23] and [24]). We found that starlings can become infected by both direct inoculation and exposure to IAV-contaminated water, but that transmission between starlings is limited. Thus, starlings may have the potential to act as IAV bridge hosts but are unlikely maintenance hosts.

## Results

### Starling experimental inoculation

We assessed the susceptibility of starlings to a low pathogenic IAV with a straightforward experimental infection study. All directly inoculated starlings became infected. Seven of the nine individuals began shedding viral RNA within 24 hours of inoculation with one individual initiating shedding on 2 days post inoculation (DPI) and one on 4 DPI. Shedding fell to near zero across all individuals by 7 DPI (Fig 1). The primary site of viral RNA excretion was the oral cavity with oral swabs showing considerably higher levels of viral RNA compared to fecal and cloacal swabs. Peak oral shedding varied by individual, with a mode of 4 DPI (range: 1–4 DPI). The mean peak concentration for oral swabs was 3.10 $\log_{10}$ EID$_{50}$ equivalents/mL (range: 2.32–3.58). While this level of shedding may appear moderate at first glance, starlings are very gregarious in the fall and winter and can form flocks in the tens of thousands, magnifying the potential pathogen load at the flock level. Cloacal shedding was mostly absent with

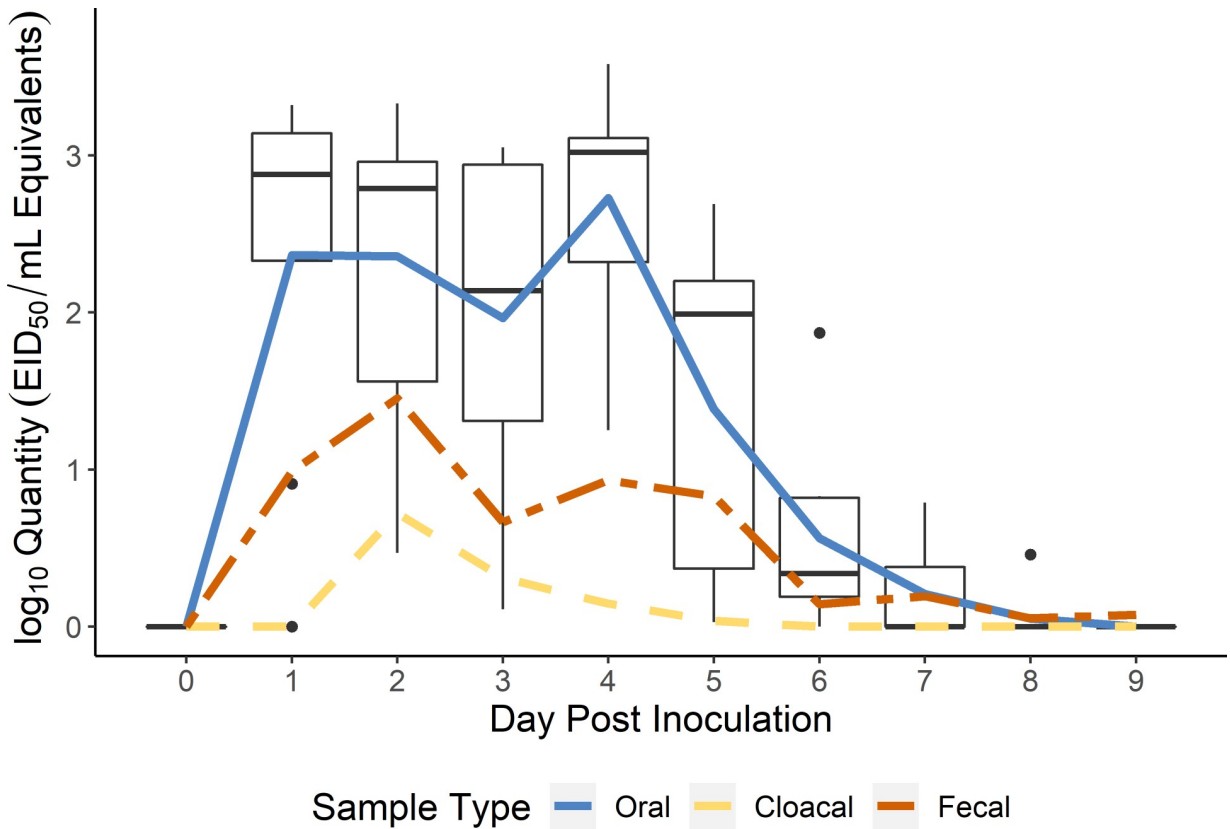

**Fig 1. Viral RNA Shedding for European Starlings Inoculated with Influenza A Virus.** Quantitative RT-PCR results for oral swabs collected from European starlings experimentally inoculated with an H4N6 influenza A virus indicate the oral cavity was the primary site of shedding. The blue (oral swabs), yellow (cloacal swabs), and red (fecal swabs) lines show mean viral RNA quantities shed (N = 9) for each sample type. The boxplot shows results for oral swabs and provides an indication of variability across individuals. For the boxplot, horizontal bars are medians, boxes outline the interquartile range, IQR, which is the range of the middle 50% of values, vertical lines are values within 1.5*IQR, and outliers are plotted as individual points.

only a single swab on 2 DPI showing a quantity greater than 2 $\log_{10}$ $EID_{50}$ equivalents/mL (Table 1). Fecal swabs showed much more variation (Table 1) with peak RNA shedding on 2 DPI with a mean of 1.5 $\log_{10}$ $EID_{50}$. Fecal shedding generally lasted through 5 DPI of the experiment.

All starlings seroconverted by 10 DPI (Fig 2). Six of the birds had detectable antibodies on 7 DPI and a single bird was positive for antibodies to IAV on 2 DPI, possibly indicating prior exposure and amnestic response. However, this individual shed viral RNA for five days and did not show evidence of prior cross immunity.

## Mallard to starling water transmission

After establishing that starlings were susceptible to an endemic wild bird IAV, we tested whether starlings could become infected by exposure to water contaminated by IAV-infected mallards (*Anas platyrhynchos*). While prior studies have evaluated contact transmission in starlings, a clear demonstration of water transmission has not been previously shown. We replicated this experiment three times using a specially constructed transmission cage designed to allow exposure to contaminated water, but no direct contact between species (Figs 3 and 4).

All of the starlings across the three transmission study replicates became infected after exposure to water contaminated by IAV-infected mallards (Fig 5). In the first replicate, eight

**Table 1. Quantitative RT-PCR results (calibrated to $EID_{50}$/mL equivalents) for European starling cloacal and fecal swabs collected during experimental inoculation, water transmission, and intraspecific transmission experiments.**

| Experiment | Sample Type | Day | 1 | 2 | 3 | 4 | 5 | 6 | 7 | 8 | 9 | 10 |
|---|---|---|---|---|---|---|---|---|---|---|---|---|
| **Experimental** | Cloacal | Mean | 0.00 | 0.72 | 0.31 | 0.15 | 0.04 | 0.00 | 0.00 | 0.00 | 0.00 | 0.00 |
| **Inoculation** | | Max | 0.00 | 2.58 | 1.45 | 1.33 | 0.33 | 0.00 | 0.00 | 0.00 | 0.00 | 0.00 |
| N = 10 | Fecal | Mean | 0.99 | 1.45 | 0.67 | 0.93 | 0.83 | 0.14 | 0.19 | 0.05 | 0.08 | 0.13 |
| | | Max | 1.83 | 3.12 | 2.13 | 1.94 | 2.00 | 0.68 | 0.72 | 0.48 | 0.68 | 0.64 |
| **Water** | Cloacal | Mean | 0.00 | 0.00 | 0.64 | 0.13 | 0.05 | 0.18 | 0.18 | 0.05 | 0.12 | 0.23 |
| **Transmission** | | Max | 0.00 | 0.00 | 1.89 | 0.73 | 0.45 | 0.87 | 1.60 | 0.44 | 0.42 | 0.79 |
| Replicate 1 | Fecal | Mean | 0.03 | 0.42 | 1.54 | 0.88 | 1.45 | 1.68 | 0.70 | 0.54 | 0.35 | 0.17 |
| N = 9 | | Max | 0.24 | 1.31 | 2.63 | 3.02 | 2.04 | 2.24 | 1.40 | 1.36 | 0.94 | 0.59 |
| **Water** | Cloacal | Mean | 0.00 | 0.30 | 0.00 | 0.00 | 0.66 | 0.60 | 0.69 | 0.16 | 0.00 | 0.22 |
| **Transmission** | | Max | 0.00 | 1.12 | 0.00 | 0.00 | 4.13 | 3.09 | 1.82 | 0.96 | 0.00 | 2.00 |
| Replicate 2 | Fecal | Mean | 0.14 | 1.06 | 1.25 | 1.48 | 0.47 | 0.63 | 0.80 | 0.67 | 0.91 | 0.35 |
| N = 9 | | Max | 1.28 | 4.07 | 3.56 | 2.67 | 1.87 | 1.88 | 2.10 | 1.55 | 1.54 | 0.82 |
| **Water** | Cloacal | Mean | 0.00 | 0.14 | 0.00 | 0.00 | 0.36 | 0.75 | 0.54 | 0.21 | 0.28 | 0.34 |
| **Transmission** | | Max | 0.00 | 1.22 | 0.00 | 0.00 | 1.94 | 2.93 | 2.50 | 1.40 | 0.97 | 1.41 |
| Replicate 3 | Fecal | Mean | 0.00 | 0.00 | 0.44 | 0.25 | 1.05 | 1.66 | 1.16 | 0.78 | 0.92 | 0.32 |
| N = 9 | | Max | 0.00 | 0.00 | 2.21 | 2.22 | 1.60 | 3.74 | 1.48 | 1.41 | 1.24 | 1.05 |
| **Intraspecific** | Cloacal | Mean | 1.34 | 0.43 | 0.39 | 0.20 | 0.35 | 0.13 | 0.04 | NA | NA | NA |
| **Transmission**[*] | | Max | 3.50 | 3.74 | 2.95 | 1.58 | 2.42 | 1.39 | 0.39 | NA | NA | NA |

[*]Naïve contacts did not become infected so these results are for directly inoculated starlings only (N = 20).

of nine individuals began shedding viral RNA orally on 2 days post contact (DPC), peak shedding occurred on 3 DPC, the mean peak concentration was 2.76 $\log_{10}$ $EID_{50}$ equivalents/mL, and most shedding ceased by 8 DPC (Fig 5A). In replicate two, oral shedding was much more variable across individuals with two starlings each initiating shedding on day 2, 3, 4, and 5 post contact and a single individual did not start shedding until 6 DPC (Fig 5B). Concomitantly, viral RNA shedding peaks were variable across individuals, ranging between 2 and 7 DPC. The mean peak concentration was 3.20 $\log_{10}$ $EID_{50}$ equivalents/mL with some individuals continuing to shed on 10 DPC. In the third replicate of the experiment, one individual initiated oral shedding on 3 DPC, five of nine individuals initiated shedding on 4 DPC, and the remaining two starlings started shedding on 5 DPC (Fig 5C). The mean peak shedding concentration was 2.93 $\log_{10}$ $EID_{50}$ equivalents/mL with the mean peak occurring on 6 DPC (range: 3–9). Similar to the experimental infection, no starlings in the water transmission experiment shed significant levels of viral RNA by either the cloacal or fecal route in any of the three replicates (Table 1).

Several starlings in replicates one and two were positive for antibodies to IAV on 10 DPC at the end of the sampling periods. The birds in replicate three were held and sampled through 59 DPC. Overall, eight of the nine starlings in that replicate mounted an antibody response. Four birds had detectable antibodies on 10 DPC and seven on 17 DPC. Only two starlings remained antibody positive by 52 DPC. Peak detection occurred on 14 DPC and a majority of the starlings were negative by 36 DPC (Fig 2).

All mallards shed viral RNA by the oral, cloacal, and fecal routes in each of the three replicates with peak shedding varying between 2–4 DPI (Fig 6). Peak concentrations were relatively consistent between replicates, with most fecal swabs peaking at about 6 $\log_{10}$ $EID_{50}$ equivalents/mL (Fig 6A–6C). In general, viral RNA concentrations in the pool reached approximately 3 $\log_{10}$ $EID_{50}$ equivalents/mL on 2 DPI and peaked at 4 $\log_{10}$ $EID_{50}$ equivalents/mL on 5 DPI.

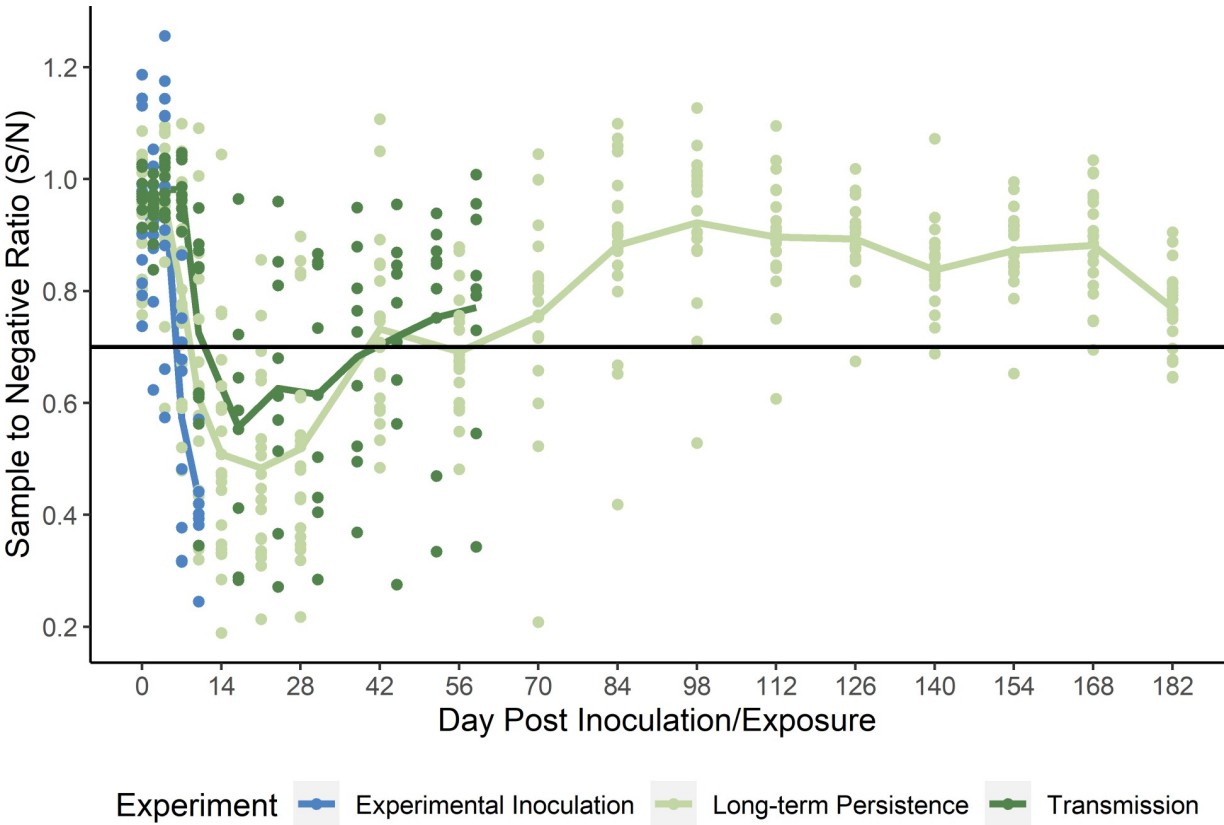

**Fig 2. Serological Results for European Starlings.** Serology results for all starlings in each of the three experiments: experimental inoculation (blue), replicate 3 of the water transmission experiment (dark green), and the antibody persistence study (light green) show a rapid waning of detectable antibodies. Values below the black horizontal line are positive for antibodies to influenza A virus by the IDEXX Multi-S bELISA using a threshold of 0.7 [47].

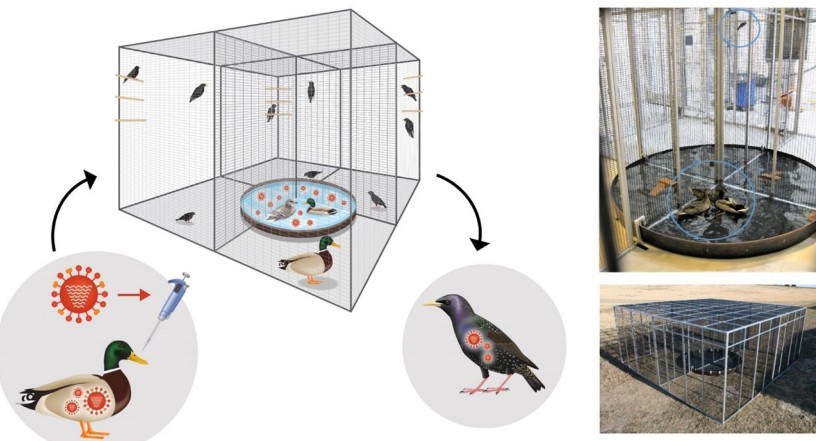

**Fig 3. Transmission Cage.** The figure shows a schematic of the transmission cage and the water transmission experiment in which we simulated natural environmental transmission by exposing starlings to water shared with mallards infected with influenza A virus (IAV). The transmission cage features four pens and a 750 L simulated pond spanning each pen, allowing shared water between species. Blue circles in the upper right photo show a starling and mallards in their separate pens.

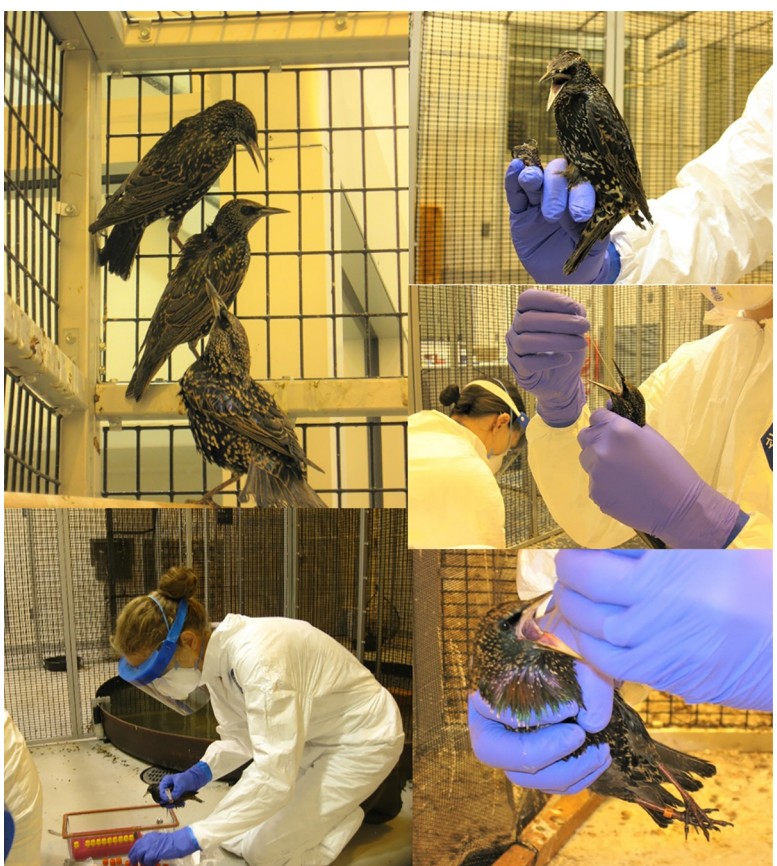

**Fig 4. Sampling Photographs.** Sampling photographs show European starlings perched (top left) or held (top right) in the transmission cage. The lower left photograph shows a researcher preparing to sample a starling and the two pictures on the lower right show oral swab collection.

The viral RNA concentration in the water remained high through the end of testing on 10 DPI.

## Intraspecific starling transmission and antibody persistence

After confirming environmental transmission in starlings sharing resources with reservoir hosts, we investigated whether starlings might act as maintenance hosts for IAVs with an intraspecific transmission study. In this experiment, all experimentally infected starlings produced positive oral swabs by 3 DPI and were negative by 7 DPI (Fig 7). Similar to the experimental inoculation study, fecal and cloacal swabs did not show significant shedding for any exposed birds (Table 1). Oral shedding peaked on 1 DPI at 3.0 $\log_{10}$ EID$_{50}$equivalents/mL. No contact starlings showed evidence of shedding viral RNA via quantitative real-time, reverse transcriptase polymerase chain reaction (qPCR) or exposure via ELISA. Long-term antibody persistence for inoculated starlings showed a positive response for starlings starting on 10 DPI with a peak on 21 DPI. Antibodies to IAV were detected in twelve of twenty individuals on 56 DPI, four individuals on 70 DPI, and one individual on 108 DPI (Fig 2).

## Discussion

The evaluation of European starlings as potential bridge or maintenance hosts of avian IAVs presented here demonstrates that starlings are 1) susceptible to an endemic North American

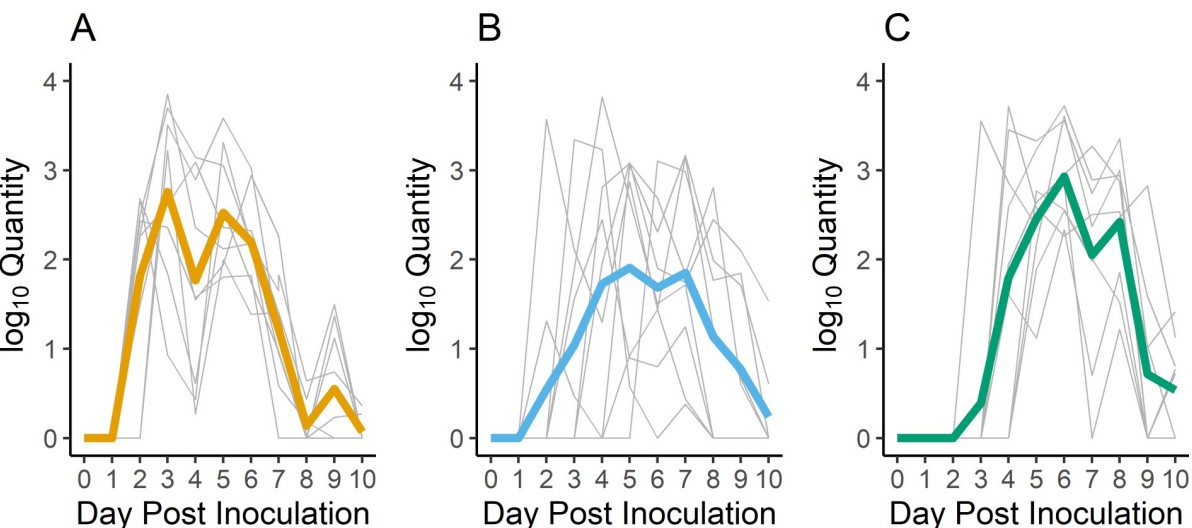

**Fig 5. Viral RNA Shedding for European Starlings Exposed to Water Contaminated with Influenza A Virus from Infected Mallards.**
Quantitative RT-PCR (qPCR) results from oral swabs across the three replicates of the water transmission experiment show all starlings became infected, but the time to infection varied across replicates. Results from the individual replicates are shown in A. replicate 1, B. replicate 2, and C. replicate 3. The qPCR results were calibrated to a known standard and results are reported as $\log_{10}$ $EID_{50}$ equivalents/mL. The thick colored lines are means across individuals and the thin gray lines are results for individual starlings.

IAV, 2) replicate viral RNA efficiently in the oral cavity, 3) can be infected when exposed to water contaminated by IAV-infected mallards, 4) do not readily transmit virus to conspecifics, and 5) exhibit a relatively brief detectable antibody response. Overall, these results indicate that while starlings are unlikely to act as maintenance or reservoir hosts for IAVs due to limited intraspecies transmission, this species may have the potential to act as a bridge host if exposed to IAVs in natural settings.

The finding that all 27 starlings across the three replicates of the transmission experiment became infected suggests that these birds are readily infected when exposed to naturally contaminated water. Interestingly, the pattern of transmission differed between the three replicates, demonstrating the importance of experimental replication in capturing variability. On the other hand, the mean infection dynamics exhibited in the experimental inoculation, environmental transmission, and intraspecies transmission studies were relatively stable. While the averages across the three studies were similar, infection dynamics across individual birds did vary, highlighting the importance of individual heterogeneity.

A number of field studies have provided evidence that wild caught European starlings can be naturally infected or exposed to IAVs [19,25–31]. However, in aggregate these studies suggest that starlings are not frequently infected or exposed to IAVs [32–34]. The relatively low seroprevalence observed in these studies might be partially explained by the relative transience of detectable antibodies demonstrated in our study. Several of the documented exposures [19,28,31] were from starlings sampled in association with poultry outbreaks, potentially supporting the idea that while starlings are not maintenance hosts for IAVs, they can act as spillover hosts.

In general, our results are in line with previous studies that have experimentally assessed IAV infection dynamics in starlings [18,29,35–39] through experimental inoculations. In general, these studies show that starlings can become infected with IAVs and seroconvert (but see [36]), primarily shed via the oral cavity (but see [29]), and exhibit limited, if any, contact transmission [35,38]. While a variety of IAV subtypes have been tested (H2, H3, H4, H5, and H7,

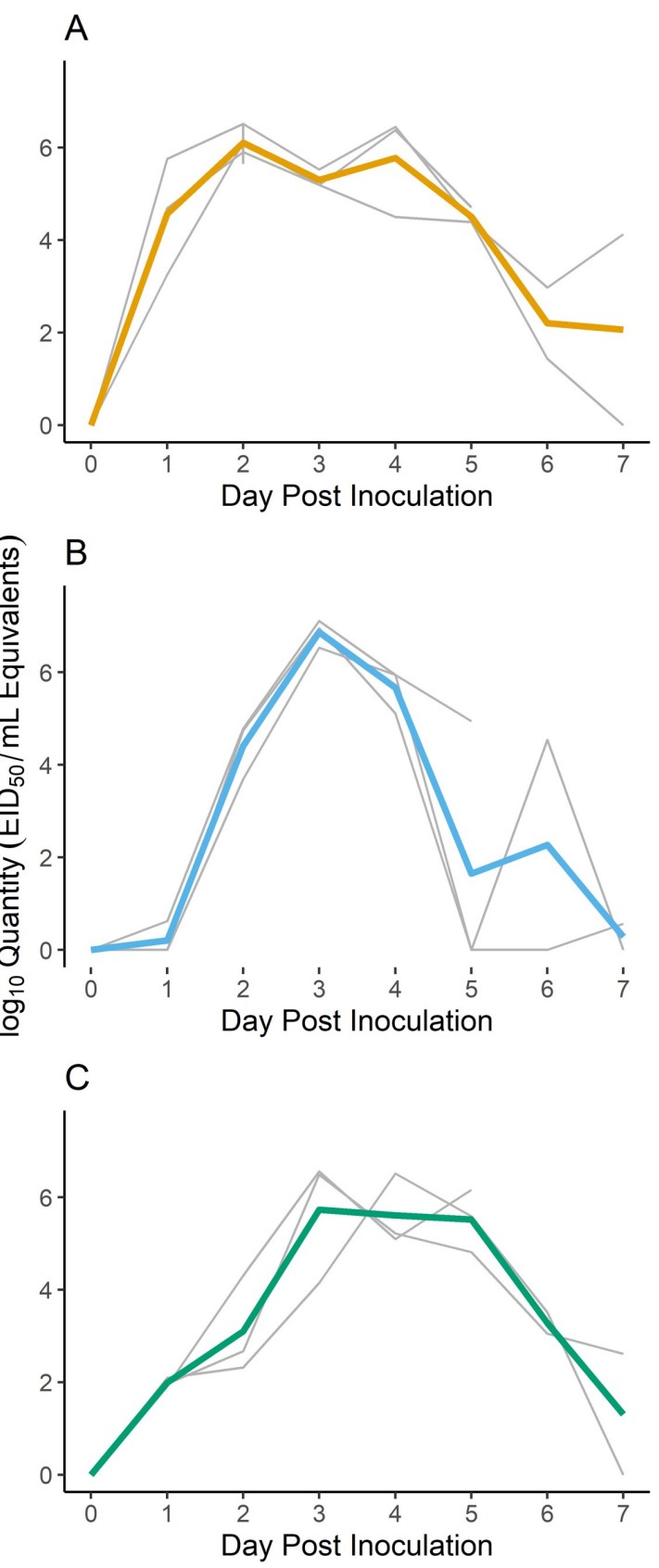

**Fig 6. Viral RNA Shedding for Mallards Inoculated with Influenza A Virus.** Quantitative RT-PCR results for mallard fecal swabs and water samples collected during the water transmission study for A. replicate 1, B. replicate 2, and C. replicate 3 show similar peaks in shedding, but variable peak days. Results from water samples collected from the shared water pool are shown in D (combined results from replicates 2 and 3). In A, B, and C, the colored lines are means across swabs and the thin gray lines show results for individual swabs.

see [18]) pathogenesis and infection characteristics have varied both within and between subtypes. Thus, infection dynamics in starlings may be strain rather than subtype specific.

Peak shedding in starlings in our experiments was above 3 $\log_{10}$ EID$_{50}$ equivalents/mL (Fig 2). Similar levels have been shown to be infectious to mallards and poultry in experimental settings [24,40–41]. Therefore, if IAV-infected starlings, attracted by food resources, nesting cavities, or roosting sites, were to enter a poultry barn they could directly or indirectly transmit IAV to poultry. Conversely, if naïve starlings came into contact with contaminated resources (e.g., food, water) at an outbreak site, they could potentially transmit the virus outside the facility (e.g., to natural areas or other poultry premises). The results of this study lay the foundation for follow-up experimental studies that test these possibilities.

Our study confirmed the ability of starlings to become infected from IAV contaminated water in a controlled environment. The concentration of viral RNA in the water pool during the water transmission replicates reached approximately 4 $\log_{10}$ EID$_{50}$ equivalents/mL in each of the three replicates. This concentration was sufficient to infect all starlings exposed to the

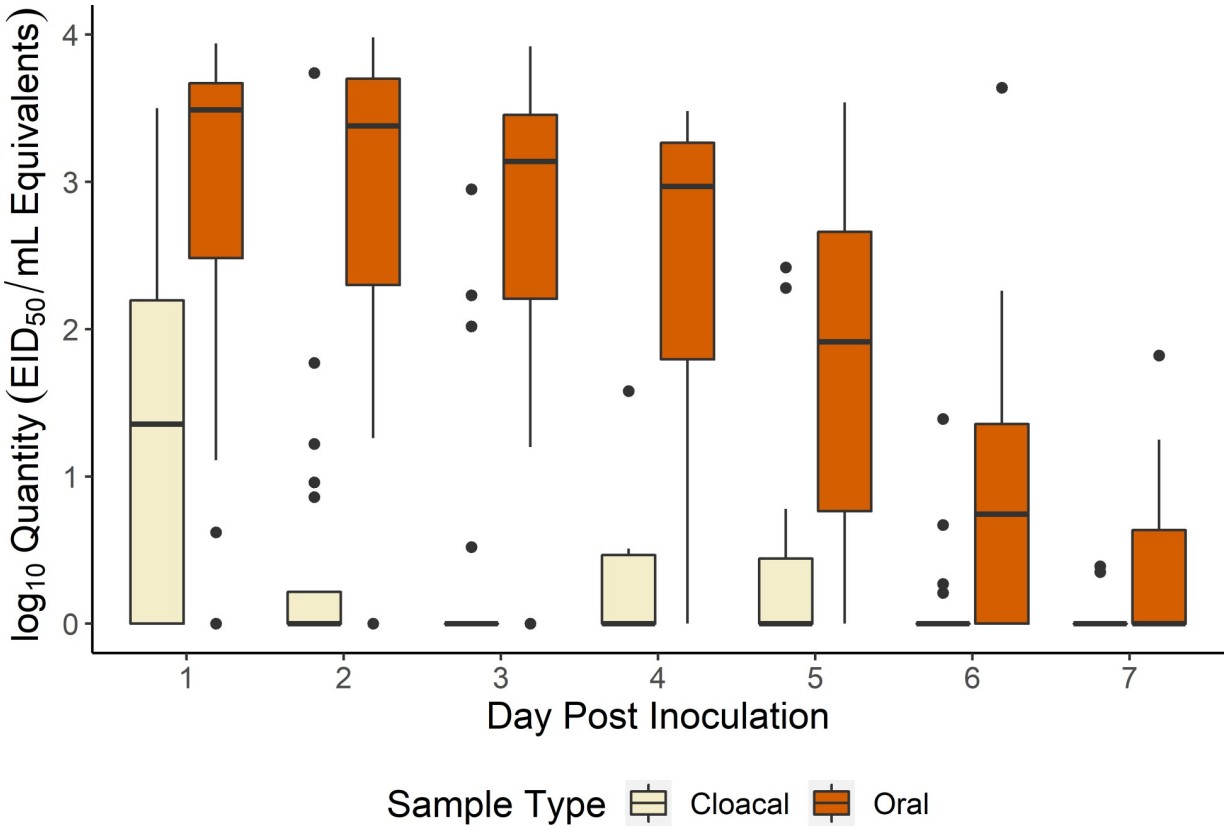

**Fig 7. Viral RNA Shedding for European Starlings Inoculated with Influenza A Virus for the Intraspecific Transmission Experiment.** Quantitative RT-PCR for oral and cloacal swabs collected from 20 inoculated starlings confirm the results of the initial experimental infection study which showed that the oral cavity is the primary site of shedding. None of the 16 contact starlings shed viral RNA.

contaminated water. Of note, we confirmed concentrations based on detection of viral RNA with qPCR, but infectious virus concentrations may have been lower and may have played a role in the lack of intra-species contact transmission.

In nature, environmental transmission depends on a variety of factors. First, the size, movement, temperature, and salinity of the water source is likely associated with transmission potential [42–44]. Small ponds or puddles that are frequently visited by IAV-infected waterfowl could contain high concentrations of virus. Second, the number of infected reservoir hosts could play an important role in mediating transmission as large flocks in which multiple birds shed virus could collectively introduce a high pathogen load [45] into the environment.

We only provided a single food dish and small poultry waterers to the starlings in the intraspecific starling transmission study. Had we provided an alternative water source such as a pool or large open bowl, we may have gotten different results. In the mallard to starling water transmission study, we anecdotally observed that starlings spent a significant amount of time in the water bathing, preening, and drinking which may have increased the likelihood of transmission. In contrast, no pool or puddles were available in the intraspecific transmission study which may have reduced the probability of transmission. Inoculated starlings did not readily transmit the virus among conspecifics, which may indicate potentially inefficient transmission to poultry. On the other hand, starlings readily acquired infections from water contaminated by mallards, suggesting multiple starlings could become simultaneously infected and jointly produce an infective dose to poultry, particularly if some species have relatively lower infectious doses compared to starlings. In the fall and winter, starlings often congregate in large flocks on or near farms and adjacent wetlands, which is potentially problematic [20] because even low infection prevalence or excretion could collectively pose a risk if host abundance is high and virus builds up in the environment [46].

The serological results from each of our experiments yielded useful information for interpreting serological data from field studies of starlings. Antibodies to IAV were detectable in most birds by 10–14 DPI or DPC but fell below the detection threshold within six weeks for half the birds and only two of twenty birds retained for long-term testing had detectable antibodies at 12 weeks post exposure. Moreover, based on prior work, we applied a less stringent threshold for a positive for the widely used ELISA used in this study [47]. Using the manufacturers recommended threshold could further decrease the window of antibody detection. Our results suggest the ideal timeframe for detecting antibodies in starlings is between 10 and 35 days post exposure. Consequently, field studies that have not found serological evidence of IAV exposure in starlings months after an outbreak [46,48] are not surprising. Timely surveillance response to an outbreak is necessary to determine if starlings may have played a role.

## Conclusions

This study shows that European starlings can contract IAV infections from direct inoculation or indirect transmission from a reservoir host through a shared water source. Shedding in this species is predominantly through the oral route, with the bulk of the shedding occurring between one and 10 DPI. Similarly, a relatively brief window of reliable antibody detection (e.g., 10 to 35 DPC) was noted and should be taken into consideration in outbreak surveillance investigations. Because IAV was readily transmitted from shedding mallards to naïve starlings via a shared water source, we suggest that water sources used by both waterfowl and starlings should be considered a possible indirect transmission mechanism for IAVs for this species. Further, the synanthropic nature of starlings and their susceptibility to multiple IAVs suggests they should be considered as a potential bridge host of concern when considering IAV trafficking risk to poultry operations. Future evaluation of transmission from IAV-infected starlings

sharing resources with poultry is a clear next step to evaluate starlings as bridge hosts for IAVs between waterfowl and poultry. Further, if transmission to poultry is confirmed, starlings will warrant further scrutiny to identify factors that may impact IAV dynamics such as other virus strains, sex, age, and immune status.

## Materials and methods

### Ethics statement

All animal procedures were approved by the Institutional Animal Care and Use Committee of the United States Department of Agriculture/Animal and Plant Health Inspection Service/Wildlife Services/National Wildlife Research Center (NWRC, Approval QA-2614), Fort Collins, CO, US. Starlings were caught and maintained under Colorado Parks and Wildlife permits 17TRb2379 and 18TRb2379.

### Overall design

This study was conducted in three parts: (1) an experimental inoculation of starlings exposed to a North American H4N6 IAV to assess susceptibility and replication kinetics, (2) an environmental transmission study to determine whether starlings can be infected by sharing water with IAV-infected mallards, and (3) an intraspecies transmission study to evaluate contact transmission from infected starlings and long-term antibody persistence.

### Animal capture and care

Starlings were wild caught in large baited drop-in traps in Weld County, Colorado, US, transferred to the NWRC campus, and then held in outdoor bird pens until testing. Day old mallards were purchased from Murray McMurray Hatchery (Webster City, IA, US), initially raised indoors, but then moved to large outdoor flight pens to await experimentation at approximately 3–5 months old. All birds were provided food and water *ad libitum* throughout the experiment and were screened for IAV viral RNA and antibodies to IAV prior to experimental testing. All mallards were negative for IAV exposure, but a few starlings showed suspect positive antibody results and were not used in the study.

During infection testing, all birds were housed in a Biosafety Level 2 (BSL-2) animal room equipped with a four-quadrant transmission cage, custom designed for experimental studies of pathogen transmission (Figs 3 and 4). The cage is subdivided into four pens and features a central 750 L experimental pond spanning each pen to simulate natural shared water. Each pen is approximately 30.8 m$^3$. Each of the pens housing starlings was equipped with two dowel rods for perching and stacked bricks in the pond to provide a platform for drinking. The pen used to house mallards included a rubber floor mat for foot relief and a ramp into the pond. Note: all experimental infections were conducted in a Biosafety Level 2 animal room and the full transmission cage is only shown outdoors for perspective (Fig 3).

### Starling experimental inoculation

We experimentally inoculated starlings with a North American wild bird IAV to assess susceptibility, the primary site of IAV replication, and shedding dynamics. We placed nine starlings in the transmission cage (three pens of two birds each and one pen with three starlings) and experimentally inoculated all individuals with a low pathogenic H4N6 avian IAV (A/Mallard/CO/P70F1-03/08 (H4N6)) originally collected from wild bird feces during avian influenza surveillance activities [49] and then passaged through a mallard [24]. We delivered the inoculum in two doses of 100 μL, each prepared with $10^5$ EID$_{50}$ of the H4N6 IAV diluted in BA-1 viral

transport media (M199-Hank's salts, 1% bovine serum albumin, 350 mg/l sodium bicarbonate, 2.5 mg/mL amphotericin B in 0.05 M Tris, 100 mg/ml penicillin, 100 mg/mL streptomycin, pH 7.6). Specifically, we delivered a single drop from a pipet to one eye and the remainder oro-choanally. We repeated the procedure approximately four hours later, applying the inoculum to the other eye as well as oro-choanally. Oral, cloacal, and fecal swabs were collected in one mL of BA-1 daily for 10 days post-inoculation (DPI). To obtain individual specific fecal swabs, starlings were placed in ventilated plastic boxes until a sample was available. Sample boxes were cleaned and disinfected each day to prepare for the next day of sampling. Swab samples were kept on ice during sampling and placed in -80° C ultra-cold freezers until laboratory testing. We collected blood by jugular and brachial venipuncture into serum separator microtubes on days 2, 4, 7 and 10 DPI. Blood samples were centrifuged at 3.5 G for 10 minutes and held at 4° C until testing. In this and subsequent experiments, we took significant measures (e.g., foot baths, changing PPE, limited entry and egress) to prevent cross-contamination between pens within the transmission cage.

### Mallard to starling water transmission

We tested whether IAV is transmitted from mallards to starlings via shared water in an environmental transmission experiment that we replicated three times. Three naïve mallards (N = 9 across three replicates) were placed in one of the four pens of the transmission cage and nine starlings (N = 27 across three replicates) were placed in the remaining three pens with three birds per pen. We oro-choanally inoculated mallards with $10^5$ EID$_{50}$ H4N6 IAV diluted in 1 mL BA-1. We collected oral and cloacal swabs from each bird (mallards and starlings) daily for 10 days. We collected individual fecal samples from starlings, but duck fecal samples were collected from the pen floor (N = 3 per day). We also collected four 1 mL water samples from the artificial pond each day (one sample per pen quadrant). On days 2, 4, 7, and 10 we collected blood from all birds by brachial, jugular (starlings), or medial metatarsal (ducks) venipuncture. Swab samples were placed in one mL BA-1, water samples were placed in 0.5 mL BA-1, and then all samples were stored at -80° C until testing. Blood was centrifuged and stored at 4° C until testing. One mallard each was euthanized and necropsied on 5, 7, or 10 DPI to harvest tissues for a separate study. All starlings were euthanized on 10 DPC with the water pool. The room was cleaned and sanitized with a 10% bleach mixture to prepare for the next experimental replicate. Following the conclusion of the third experimental transmission replicate, the nine starlings were held until 59 DPC and blood was collected weekly to characterise antibody kinetics.

### Intraspecific starling transmission and long-term antibody persistence

Based on the results of the water transmission study, we conducted a third experiment to test intraspecific transmission to naïve contact starlings and long-term antibody dynamics after a known exposure time point. Nine starlings were placed in each of the four pens of the transmission cage. We inoculated five birds per pen (N = 20) as previously described and the four remaining starlings per pen served as naïve contacts (N = 16). We collected oral and cloacal swabs daily through 7 DPI for inoculated birds and 10 DPC for contact birds. We collected blood from all birds on days 4, 7, 10, 14, 21, 28, 42, and 55. Contact birds were euthanized on 56 DPC. We continued blood collection from inoculated birds every two weeks for approximately six months.

### Laboratory analyses

Water samples and oral, cloacal, and fecal swabs were tested by qPCR. Viral RNA was extracted using MagMax-96 AI/ND Viral RNA Isolation Kits (Thermo Fisher Scientific, Inc.,

Waltham, MA). Duplicate RNA extracts were tested using primers and a probe specific for the influenza type A matrix gene [50] using Bio-Rad iTaq Universal Probes One-Step Kits and Bio-Rad CFX96 Touch Thermocyclers (Bio-Rad Laboratories, Inc., Hercules, CA). Thermocycler conditions followed those previously described [51] except plates were run for 40 cycles of 95˚C for 15 seconds and 60˚C for 30 seconds. H1N1 IAV calibrators diluted to viral titres of $10^2$, $10^3$, $10^4$, and $10^5$ $EID_{50}$/mL were tested in duplicate on each plate and used to construct four-point standard curves. Sample viral RNA quantities were extrapolated from the standard curves and are reported as PCR $EID_{50}$ equivalents/mL. Cycle quantities (Cq) were standardised by setting the baseline to a uniform threshold across all runs.

We tested serum samples for antibodies reactive to IAV by enzyme-linked immunosorbent assay (ELISA) using the FlockCheck Avian Influenza MultiS-Screen Antibody Test Kit (IDEXX Laboratories, Inc., Westbrook, ME) following manufacturer's instructions except we used a classification threshold of 0.7 sample-to-negative (S/N) ratio [47,52].

## Supporting information

**S1 Data. European starling experimental infection qPCR data (Fig 1 and Table 1).** These data are associated with Fig 1 and Table 1 in the manuscript. Column headers are as follows: TYPE = sample type collected. ORAL = oral swab, CLOACAL = cloacal swab, FECAL = fecal swab. Note fecal swabs were collected from pen floors and were not associated with a particular individual. BAND = unique leg band associated with each individual bird. DPI = day post inoculation. MeanQTY = viral RNA concentration based on calibrated $EID_{50}$/mL equivalents. Value is the mean of duplicate qPCR wells. MeanQTY+1 = MeanQTY + 1. Log(MeanQTY+1) = log based 10 of MeanQTY+1.
(CSV)

**S2 Data. European starling serology data (Fig 2).** These data are associated with Fig 2 in the manuscript. Column headers are as follows: EXPERIMENT = the experiment the data are derived from. Experimental Infection, Antibody Persistence, or Transmission Replicate 3. BAND = unique leg band associated with each individual bird. DPI = day post inoculation or exposure. MeanSN = Sample to Negative Ratio (SN) from the IDEXX Multi-S Assay. SN is the mean of two wells.
(CSV)

**S3 Data. European starling environmental transmission qPCR data (Fig 5 and Table 1).** These data are associated with Fig 5 and Table 1 in the manuscript. Column headers are as follows: REPLICATE = result associated with Replicate 1, 2, or 3. SPECIES = starling. BAND = unique leg band associated with each individual bird. DPI = day post exposure. MeanQTY = viral RNA concentration based on calibrated $EID_{50}$/mL equivalents. Value is the mean of duplicate qPCR wells. MeanQTY+1 = MeanQTY + 1. Log(MeanQTY+1) = log based 10 of MeanQTY+1.
(CSV)

**S4 Data. Mallard environmental transmission qPCR data (Fig 6).** These data are associated with Fig 6 in the manuscript. Column headers are as follows: REPLICATE = result associated with Replicate 1, 2, or 3. SPECIES = mallard. BAND = unique leg band associated with each individual bird or sample number from the pen floor. DPI = day post inoculation. MeanQTY = viral RNA concentration based on calibrated $EID_{50}$/mL equivalents. Value is the mean of duplicate qPCR wells. MeanQTY+1 = MeanQTY. Log(MeanQTY+1) = log based 10 of MeanQTY+1.
(CSV)

**S5 Data. Water environmental transmission qPCR data.** These data are qPCR values for water samples collected for Replicates 2 and 3 of the environmental transmission experiment.. Column headers are as follows: REPLICATE = result associated with Replicate 2, or 3. PEN = water collected from the pool associated with Pen 1, 2, 3, or 4. DPI = day post inoculation of the mallards. MeanQTY = viral RNA concentration based on calibrated $EID_{50}$/mL equivalents. Value is the mean of duplicate qPCR wells. MeanQTY+1 = MeanQTY + 1. Log (MeanQTY+1) = log based 10 of MeanQTY+1.
(CSV)

**S6 Data. European starling experimental infection qPCR data (2nd experiment for long-term antibody persistence, Fig 7).** These data are associated with Fig 7 in the manuscript. Column headers are as follows: TREATMENT = inoculated (all contact birds were negative and are not included). BAND = unique leg band associated with each individual. DPI = Day post inoculation. TYPE = sample type collected. ORAL = oral swab, CLOACAL = cloacal swab. MeanQTY = viral RNA concentration based on calibrated $EID_{50}$/mL equivalents. Value is the mean of duplicate qPCR wells. MeanQTY+1 = MeanQTY + 1. Log(MeanQTY+1) = log based 10 of MeanQTY+1.
(CSV)

## Acknowledgments

We thank Hailey McLean and Nicholas Dannemiller for animal sampling and laboratory assistance. We also thank the Animal Care staff for animal husbandry. The manuscript was reviewed for general policy statements committing the USDA to action, but the findings were independently developed by the authors.

## Author Contributions

**Conceptualization:** Jeremy W. Ellis, J. Jeffrey Root, Susan A. Shriner.

**Data curation:** Jeremy W. Ellis, Katherine L. Dirsmith, Susan A. Shriner.

**Formal analysis:** Jeremy W. Ellis, Susan A. Shriner.

**Investigation:** Jeremy W. Ellis, J. Jeffrey Root, Loredana M. McCurdy, Kevin T. Bentler, Nicole L. Barrett, Kaci K. VanDalen, Katherine L. Dirsmith, Susan A. Shriner.

**Methodology:** Jeremy W. Ellis, Susan A. Shriner.

**Project administration:** Jeremy W. Ellis, Susan A. Shriner.

**Visualization:** Susan A. Shriner.

**Writing – original draft:** Jeremy W. Ellis, Susan A. Shriner.

**Writing – review & editing:** Jeremy W. Ellis, J. Jeffrey Root, Loredana M. McCurdy, Kevin T. Bentler, Nicole L. Barrett, Kaci K. VanDalen, Katherine L. Dirsmith, Susan A. Shriner.

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
