## [Decision Letter · Decision Letter 0]

19 Feb 2021

Dear Dr. Shriner,

Thank you very much for submitting your manuscript "Avian influenza A virus susceptibility, infection, transmission, and antibody kinetics in European starlings" for consideration at PLOS Pathogens. As with all papers reviewed by the journal, your manuscript was reviewed by members of the editorial board and by several independent reviewers. In light of the reviews (below this email), we would like to invite the resubmission of a significantly-revised version that takes into account the reviewers' comments.

We cannot make any decision about publication until we have seen the revised manuscript and your response to the reviewers' comments. Your revised manuscript is also likely to be sent to reviewers for further evaluation.

Sincerely,

Daniel R. Perez, PhD

Associate Editor

PLOS Pathogens

Adolfo García-Sastre

Section Editor

PLOS Pathogens

Kasturi Haldar

Editor-in-Chief

PLOS Pathogens

orcid.org/0000-0001-5065-158X

Michael Malim

Editor-in-Chief

PLOS Pathogens

orcid.org/0000-0002-7699-2064

Reviewer's Responses to Questions

**Part I - Summary**

Reviewer #1: The work by Ellis et al. describes the experimental infection, interspecies (mallard to starling through shared water) and intraspecies (contact transmission between starlings) experiments in order to gain more information about the possibility of starlings to act as bridge hosts between reservoir species (wild waterfowl) and spill-over hosts (poultry). Although there is still limited information about the role of passerine birds in influenza virus ecology, I think that the present study does not bring sufficient added novel information compared to the experimental infection studies cited and also nicely reviewed recently by the authors (reference 20) to justify a publication in Plos Pathogens. In previous studies, the infection dynamics, antibody profile, and contact intraspecies transmission have already been described in starlings inoculated with different low pathogenic avian influenza viruses (LPAIV) and highly pathogenic avian influenza viruses (HPAIV), including an H4 virus, which is the subtype of the virus in the present study. The added value of the present work is the transmission experiment between mallards and starlings, which show that starlings can get infected via contaminated water. However, information about how the experimental set-up was designed to prevent cross-contamination between the different groups is lacking (please see below for more detailed questions about that specific point) and it makes it difficult to assess how reliable the results of this experiment are. Additionally, the authors conclude quite firmly that “starlings have the potential to act as IAV bridge hosts”, while only one “side” of the bridge was studied in this study (the mallard to starling transmission but not the starling to poultry transmission). Given the fact that there was no contact transmission observed between starlings and very limited cloacal/fecal shedding, it is very well possible that transmission from starling to poultry would also be inefficient, making starlings dead end hosts rather than bridge hosts. I think that this point should be discussed more in depth in the manuscript to balance the conclusions of the study.

Reviewer #2: The paper by Ellis and colleagues reports on a series of experiments designed to shed light on the role of European starlings in influenza a virus transmission at the wildlife -domestic bird interface. The authors use a local low pathogenic influenza virus (LPIAV) in three experiments that investigate susceptibility to infection and shedding, infection by IAV contaminated water and intraspecies transmission in the species. In addition, one experimental group was employed to test for antibody persistence over time. The experimental methods are sound, and experiments follow a logical chronologic order and tackle key basic questions with view to the potentials of the species as bridge or reservoir host for avian influenza virus. The study is outlined clearly, and results are informative and discussed in some detail.

I have a few concerns and questions regarding the study design, and interpretation of the results.

1- Could the decision to use an LP H4N6, as performing similar experiments in a BSL-3 facility is much more costly and complicated, and as it may be close in dynamics to field situations to H5/H7LPIAV have influenced some of the study results, such as antibody titres and `persistence and shedding?

2- Why have the authors used detection of viral RNA, expressed as EID50 equivalents/ml as proxy for shedding? With view to the potential of starlings as bridge hosts it would be more interesting to obtain the amount of infectious virus shed by the birds via different routes. Shedding of RNA does not necessarily imply that infectious virus is shed. This could also explain the lack of intraspecific transmission in the third experiment.

3- Were all starlings used in the experiments caught at the same time, and if so, at which time of the year? From the pictures (plumage) it appears that no juveniles (hatch-year) were among the birds used in the experiment. Would the authors expect a different outcome with juveniles?

**Part II – Major Issues: Key Experiments Required for Acceptance**

Reviewer #1: - What was the procedure to ensure that there was no cross-contamination of the naïve starlings in the mallard/starling and starling/starling transmission experiments by the researcher themselves walking from one pen to another? Were researchers exchanging PPE (including shoes or cover shoes to prevent contamination through feces) in between the groups? Were the naïve birds present during the inoculation of the donor animals or were they placed some time after the inoculation to prevent cross-contamination at the moment of the inoculation?

- All virus titers were based on extrapolation of RT-qPCR results to EID titers based on a standard curve performed with another virus (an H1N1 avian influenza virus). This approach has unfortunately several limitations. First, if this approach is chosen, I think that the same virus should be chosen to perform the standard curve as DI formation can vary between viruses. Second, the different numbers of DIs in the virus stock used as a standard and the swab samples, the detection of RNA from non-infectious viruses and dead infected cells will skew the standard curve. I think that this study would benefit from the determination of actual EID titers by titrating at least some samples in eggs.

- The choice of the virus was not motivated based on what subtype would be the most relevant to study, but on ease to work with lower containment viruses. This comment probably stems from my lack of knowledge on the biocontainment regulation in the US, but is it the case that LPAIV H5 and H7 have to be handled at different biocontainment than LPAIV H4 viruses? Otherwise, studying a LPAIV virus that is more relevant for poultry (H5, H7 or. H6) would have been more interesting.

Reviewer #2: Not applicable

**Part III – Minor Issues: Editorial and Data Presentation Modifications**

Reviewer #1: - It is not clear from the description in the material and methods at what biocontainment level were the experiments performed. In the picture in figure 3, the water transmission set-up appears to be placed outdoor. However, in figure 4, the set-up is indoor.

- How long before the start of the experiment were the birds tested for sero-negativity? How long were the starling kept outdoor in between the time they were catch from the wild and the experiment?

- “While there is no evidence….similar to low pathogenic H5/H7 strains”. A reference should be added here to support this statement.

- Results: “The primary site of viral RNA excretion was the oral cavity with oral swabs showing significantly higher levels…”. No statistical analysis was conducted to support this statement.

- Figure 1: the variation across individuals for the fecal and cloacal shedding should be also depicted.

- Inoculation of the starlings: it would be useful to understand why it was chosen to inoculated the starling via the eyes, in addition to the choanal route. After 4 hours, was only the eye inoculation repeated in the other eye, or was the choanal inoculation also repeated?

- Figure 5 and 6: data from the different groups should not be combined in one box plot graph.

- Author summary: “highly pathogenic to poultry, wild birds and humans. In fact, the last three major pandemics were caused by IAVs with genes derived by bird viruses”. It seems that here the authors refer to two different kind of evolution of avian influenza viruses, but yet link in these two sentences as if they were related, which is not the case. The first one is the transition from low to highly pathogenic avian influenza in poultry. These viruses are highly pathogenic in poultry but not necessarily in wild birds or humans. The second is reassortment between two influenza A viruses.

- Introduction: in the sentence where ref 19 is cited, it would be good to indicate the subtype of the influenza virus that was detected in the starling.

- Figure 2 could be split in different panels to increase readability.

Reviewer #2: Introduction

Page 10: Please provide a reference for this assumption “While there is no evidence that H4 IAVs cause a significant threat to wildlife, poultry, or human health, the infection kinetics of these viruses are generally similar to low pathogenic H5/H7

strains.”

PLOS authors have the option to publish the peer review history of their article (what does this mean?). If published, this will include your full peer review and any attached files.

Reviewer #1: No

Reviewer #2: No
---

## [Editor Report · Decision Letter 1]

9 Aug 2021

Dear Dr. Shriner,

We are pleased to inform you that your manuscript 'Avian influenza A virus susceptibility, infection, transmission, and antibody kinetics in European starlings' has been provisionally accepted for publication in PLOS Pathogens.

Best regards,

Daniel R. Perez, PhD

Associate Editor

PLOS Pathogens

Adolfo García-Sastre

Section Editor

PLOS Pathogens

Kasturi Haldar

Editor-in-Chief

PLOS Pathogens

orcid.org/0000-0001-5065-158X

Michael Malim

Editor-in-Chief

PLOS Pathogens

orcid.org/0000-0002-7699-2064
---

## [Editor Report · Acceptance letter]

24 Aug 2021

Dear Dr. Shriner,

We are delighted to inform you that your manuscript, "Avian influenza A virus susceptibility, infection, transmission, and antibody kinetics in European starlings," has been formally accepted for publication in PLOS Pathogens.

Best regards,

Kasturi Haldar

Editor-in-Chief

PLOS Pathogens

orcid.org/0000-0001-5065-158X

Michael Malim

Editor-in-Chief

PLOS Pathogens

orcid.org/0000-0002-7699-2064